# Graphene Oxide Layer-by-Layer Films for Sensors and Devices

**DOI:** 10.3390/nano11061556

**Published:** 2021-06-12

**Authors:** Ivan C. C. Assunção, Susana Sério, Quirina Ferreira, Nykola C. Jones, Søren V. Hoffmann, Paulo A. Ribeiro, Maria Raposo

**Affiliations:** 1Laboratory of Instrumentation, Biomedical Engineering and Radiation Physics (LIBPhys-UNL), Department of Physics, NOVA School of Science and Technology, NOVA University Lisbon, 2829-516 Caparica, Portugal; i.assuncao@campus.fct.unl.pt (I.C.C.A.); susana.serio@fct.unl.pt (S.S.); 2Instituto de Telecomunicações, Avenida Rovisco Pais, 1, 1049-001 Lisboa, Portugal; quirinatf@gmail.com; 3ISA, Department of Physics and Astronomy, Aarhus University, Ny Munkegade 120, 8000 Aarhus C, Denmark; nykj@phys.au.dk (N.C.J.); vronning@phys.au.dk (S.V.H.)

**Keywords:** graphene oxide, layer-by-layer films, sensors, solar cells, vacuum ultraviolet

## Abstract

Layer-by-layer films of poly (allylamine hydrochloride) (PAH) and graphene oxide (GO) were characterized, looking at growth with the number of bilayers, morphology, and electrical properties. The PAH/GO films revealed a linear increase in absorbance with the increase in the number of deposited bilayers, allowing the determination that 10.7 ± 0.1 mg m^−2^ of GO is adsorbed per unit of area of each bilayer. GO absorption bands at 146, 210, 247 and 299 nm, assigned to π-π* and n-π* transitions in the aromatic ring (phenol) and of the carboxylic group, respectively, were characterized by vacuum ultraviolet spectroscopy. The morphological characterization of these films demonstrated that they are not completely uniform, with a bilayer thickness of 10.5 ± 0.7 nm. This study also revealed that the films are composed of GO and/or PAH/GO fibers and that GO is completely adsorbed on top of PAH. The electrical properties of the films reveal that PAH/GO films present a semiconductor behavior. In addition, a slight decrease in conduction was observed when films were prepared in the presence of visible light, likely due to the presence of oxygen and moisture that contributes to the damage of GO molecules.

## 1. Introduction

In recent years there has been a revolution in materials science research, with the successive discoveries of allotropic carbon nanomaterials such as fullerenes, carbon nanotubes and graphene, due to the promising properties that these materials can present. They have exceptional characteristics for science and technology, as they are non-toxic, chemically and thermally tolerant, and mechanically robust [1]. Graphene, for example, has a molecular structure that is composed of carbon atoms organized in a regular, two-dimensional (2D) hexagonal pattern, with the hybridized sp^2^ carbon atoms bonded to each other. Graphene, with a thickness of only one atom, ~0.34 nm, basically consists of a single isolated layer of graphite and is considered to be the basis for the construction of all carbon allotropes, such as fullerene (0D), carbon nanotubes (1D) and graphite (3D) [2]. Due to the two-dimensional structure, the quantum confinement of the electrons gives graphene new and unique properties, which is why it has attracted more and more attention from the scientific community. Graphene has excellent structural, physical-chemical, mechanical, electrical, and optoelectronic properties [1,2,3,4,5]

Graphene applications now cover a range of energy storage systems to efficiently convert and store energy, such as lithium-ion batteries, solar cells, supercapacitors, among other applications [4]. Another reason for the significant interest in graphene comes from its potential use in low cost, flexible, high efficiency photovoltaic systems [1,4]. Recent studies have revealed two types of graphene-based solar cells that have shown good performance [6,7,8]. Graphene thin films have been used as a transparent electrode for dye-sensitized solar cells, and organic photovoltaic devices make use of a single graphene layer as a charge carrier with a load mobility of the order of ~2 × 10^5^ cm^2^ V s^−1^, about 200 times greater than that of silicon; this makes it an excellent alternative for transport layers and separation of charge in solar cells [2].

As research on graphene has developed, several engineering applications have also emerged, such as nanostructured materials, conductive polymers, battery electrodes, and so on. These applications resulted in the production not only of graphene monolayers but also of related materials such as graphene nano-sheets, graphene oxide, and reduced graphene oxide. As with carbon nanotubes, which vary in the number of walls, diameter, length or surface chemistry, the graphene family of materials also vary in the number of layers, side dimensions, surface chemistry or even density and quality of the individual graphene sheets. It should also be noted that there are advanced modelling methods which successfully guide and assist the synthesis of such carbon-based thin and ultrathin films. Specially developed DFT theoretical approaches, such as the synthetic (growth) approach, have been applied to novel carbon-based thin films with inherent nanostructure, to aid the understanding of structural and assembly/synthesis/surface chemistry issues on these and other similar thin film materials [9,10].

Graphene oxide (GO) has a thickness of one atom and is formed from the oxidation of chemically modified graphene, produced by the extensive oxidation of crystalline graphite or graphite oxide, followed by sonication or other dispersion methods to produce a monolayer material, usually in an aqueous suspension [11]. Stankovich et al., in 2006, were able to demonstrate that an ultrasound treatment of graphite oxide in water resulted in its exfoliation, forming stable aqueous dispersions which consist almost entirely of GO sheets [12]. Since the discovery of GO, several models of structures have been proposed, with the Lerf-Klinowski configuration being the most accepted model. This model consists of non-oxidized aromatic regions and aliphatic six-membered rings containing hydroxyl (OH), epoxy (-O-) groups and isolated C=C bonds, and having OH, carbonyl (C=O) and carboxyl groups (COOH) at the ends [10]. The presence of these functional groups makes the GO hydrophilic, which enables it to disperse in water. Basal planes also include non-modified graphene domains that are hydrophobic and capable of π-π interactions, which is important for the adsorption of dye or other molecules, resulting in a kind of surfactant capable of stabilizing hydrophobic molecules in solution. Samples of GO can not only contain monolayers, but also multilayer flakes, so for applications requiring monolayers, centrifugation is typically employed to remove material that is not completely exfoliated into single-thickness sheets of one atom.

GO is electrically insulating, with a resistance per area (resistance sheet) of ~10–12 Ω.m^−^² or higher, as a result of disruption of π-electron delocalized conjugation pathways by sp^3^ (C-O) bonds and hole defects. The increase of sp^3^ C-O bonds leads to an increase in the GO bandgap. Due to the complete removal of π electrons, GO has a bandgap of 3.1 eV. With this, GO can go from insulator to semiconductor and semi-metal graphene-type, by reducing or restoring sp^2^ C-C bonds. This adjustable conductivity coupled with the bandgap are two quite attractive features in the electronics and optoelectronic devices manufacturing industries.

Recently, GO-containing films were successfully used to develop sensor devices for 5-chloro-2-(2,4-dichlorophenoxy)-phenol, also designated as triclosan (TCS). TCS is an antibacterial agent widely used in soaps, toothpastes and first-aid products, which presents several problems related to its noxious effects on biological systems. Furthermore, due to its large-scale consumption, TCS is now accumulating in bodies of water [13]. The sensitive part of TCS sensor devices consists of thin films prepared by alternating adsorption of polyelectrolytes poly(ethyleneimine) (PEI) and chitosan (Chi), using a variant of the layer-by-layer (LbL) method through aerosol spray [13]. The development of thin film-based devices generally requires the use of layers of different materials to gather the desired features (e.g., electrical, optical, magnetic) or to control the sensitive layer thicknesses—an important issue to attain the necessary characteristics. The need to obtain well-organized molecular structures with accurate thickness control led to the development of new multilayer thin film manufacturing methods [14]. In the 1990s, Decher et al. [15] and references therein developed the LbL technique, which is based on the electrostatic interaction between layers of molecules with opposite charges. The multilayer formation can be related, for example, to electrostatic interactions, hydrogen bonds, donor/acceptor interactions, and the propensity of oppositely charged polyelectrolyte pairs to form complexes. The LbL technique is quite versatile, since heterostructures with different materials and specific functionalities can be produced using any molecules or species with electric charges, including, for example, polyelectrolytes, conducting polymers, proteins, DNA or dendrimers [16]. Studies have shown that LbL films have the same properties after deposition of the first layers on different types of substrates, regardless of the size and topology of the substrate. One of the most useful advantages of this technique is that films can be deposited on a specific substrate required for different characterization techniques, without altering the properties of the films. Other advantages of using this method lie in the fact that the procedure is simple, low cost and compatible with large-scale production. Another relevant detail of LbL films is the linear increase in the thickness or quantity of adsorbed material per unit area, as a function of the number of adsorbed bilayers, despite the nonlinearity that takes place during the buildup of the first few bilayers, which is attributed to the interactions between the substrate and polyelectrolytes. Moreover, the LbL film deposition method is carried out using aqueous solutions, which from an environmental point of view makes this technique lower risk. In recent years, several LbL films based on GO have been prepared and characterized, having the development of devices and sensors in mind [17,18,19,20,21,22,23].

In this article, LbL films prepared with GO and the PAH polyeletrolyte were fully characterized with the main objective of developing devices as solar cells and sensors. For such characterization, vacuum ultraviolet (VUV), UV-visible, infrared and impedance spectroscopies, electrical measurements and microscopies were used.

## 2. Materials and Methods

### 2.1. Thin Film Preparation

Thin films of the polyelectrolyte poly (allylamine hydrochloride) (PAH) (average Mw = 50,000–65,000 g/mol, CAS number 71550-12-4) and graphene oxide (GO) (CAS number 763705), both purchased from Sigma-Aldrich (Steinheim, Germany), were prepared by the layer-by-layer technique [14]. This technique consists of the alternate adsorption of cationic and anionic polyelectrolytes on solid substrates. For the present work, aqueous solutions of the PAH polyelectrolyte were prepared with a concentration of 10^−2^ M by dissolving 0.0234 g of PAH granules in 25 mL of ultra-pure water (Milli-Q, Merck KGaA, Darmstadt, Germany, with a resistivity of 18.2 MΩ.cm at 25 °C. This water was used in all solutions, and also as a washing solution. The GO solutions were obtained by dissolving 2 mL of a 2 mg/mL aqueous solution of GO in 23 mL of ultra-pure water.

In the preparation of the PAH/GO LbL films, the solid substrates were first immersed in the PAH aqueous solution for 1 min, thereby adsorbing a positively charged layer. After this adsorption, the substrate was washed with ultra-pure water (washing solution) to remove excess molecules or aggregates that had not been completely adsorbed. Subsequently, the substrate was immersed in the anionic GO aqueous solution, also for 1 min. The adsorption times were chosen based on earlier studies of adsorption kinetics and preliminary studies carried out [24]. After adsorption of the anionic layer, the substrate was again washed with ultra-pure water to remove any excess, and was then dried with a stream of nitrogen (99% purity, Air Liquide, Algés, Portugal). This additional step to the LbL process was necessary as it was found that drying with nitrogen was critical for obtaining the desired films; studies performed without drying with nitrogen at the end of each bilayer have shown that after a few bilayers (~5), the film begins to desorb and the solution of GO starts to precipitate. After the aforementioned sequence, a film with a bilayer was obtained and the procedure was repeated according to the number of bilayers, n, required. From now on, LbL films are designated as (PAH/GO)_n_.

Cast films were also prepared by dispensing (with the aid of a micropipette) some drops of the solution of the polyelectrolyte under study onto the substrate. The solvent was then removed by evaporation at room temperature, thereby forming a thin film.

For the characterization of thin films produced by the different techniques, several types of substrates were employed, namely, BK7 optical glass, quartz, calcium fluoride (CaF_2_), glass with a film of fluorine-doped tin oxide (FTO) and glass substrates with interdigitated gold electrodes (DropSens DRP-G-IDEAU5, Oviedo, Asturias, Spain.). Apart from the last three, all substrates were subjected to a cleaning process before being used in film preparation. The cleaning of the substrates was achieved by immersing them in a solution of hydrogen peroxide (H_2_O_2_) and sulfuric acid (H_2_SO_4_), in a ratio of 1:1. This solution is called the “piranha” solution, and its preparation and handling requires great care, since the reaction is exothermic. After treatment with the piranha solution, the substrates were exhaustively washed with ultra-pure water and dried with a flow of nitrogen before being used. The calcium fluoride and FTO glass substrates were sequentially washed with isopropanol, acetone, isopropanol, and ultra-pure water, and then dried with a flow of nitrogen just before use. These procedures, besides guaranteeing the cleaning of the substrate, also provide them with the negative charge required for adsorption of the PAH (positively charged) layer.

For the development of the solar cell devices, two different approaches were used for the deposition of an aluminum electrode: DC magnetron sputtering and vacuum thermal evaporation. In the first procedure, an aluminum disc, with a diameter of 64.5 mm, a thickness of 4 mm and a purity of 99.99% (Goodfellow, Huntingdon, UK), was used as a sputtering target. A turbomolecular pump was used to achieve a base pressure of 10^−5^ Pa, before introducing the argon. The total pressure during the deposition was kept constant at 0.8 Pa; the sputtering power was 130 W and the deposition time was 5 min. Before the sputter-deposition step of producing the films, a movable shutter was interposed between the target and the substrate, and the target was pre-sputtered in an argon atmosphere for 2 min to clean the target surface. The target-to-substrate distance was kept constant at 100 mm. For thermal evaporation, Al electrode deposition was carried out with Al wires of 99.5% purity from Advent Research Materials (Oxford, UK), which were placed on a spiral tungsten resistor in a vacuum chamber, with the films mounted on a vertical support at about 100 mm from the tungsten resistor. The deposition was carried out at pressures of 10^−5^–10^−6^ mbar, achieved with a turbomolecular pump, an applied current of 50 A, and a deposition time of 1 min.

### 2.2. Characterization Techniques

The films were characterized by ultraviolet-visible spectroscopy using a Shimadzu spectrophotometer (UV-2101PC) (Shimadzu Europa GmbH, Duisburg, Germany) scanning from 200 to 800 nm, and by vacuum ultraviolet (VUV) spectroscopy in the range of 125 to 340 nm, using the AU-UV beamline on the ASTRID2 synchrotron radiation source at Aarhus University, Denmark [25,26,27]. The beamline wavelength was selected by means of a toroidal dispersion grating placed upstream of the sample, with the monochromator producing photons with a resolution of 0.08 nm. Fourier transform infrared (FTIR) absorption spectra were collected in the wavenumber range of 400 to 4000 cm^−1^, with a resolution of 4 cm^−1^, using a Tensor 27 FTIR spectrometer (Bruker, Rosenheim, Germany).

Surface morphology and the thickness of films were examined using a field-emission scanning electron microscope (FEG-SEM JEOL 7001F, JEOL (Europe), SAS, Croissy-sur-Seine, France) operating at 15 keV. To prevent charge build up, a thin layer of chromium was coated on the surface of the film before analysis. The surface of the film was also examined by Atomic Force Microscopy (AFM) obtained with a Scanning Tunneling Microscopy (STM), (Agilent Technologies, model PicoScan, Santa Clara, CA, USA), and by optical microscopy using a Nikon optical microscope (model Eclipse LV100) (Nikon Metrology GmbH, Düsseldorf, Germany). All AFM images were acquired with an area of 2 × 2 µm^2^, and the surface morphology of the films was characterized by the root mean square roughness (Sq), which was calculated using Gwyddion software (SourceForge, Czech Metrology Institute, Jihlava, Czech Republic) [28].

Electrical characterization of the irradiated solutions was carried out measuring impedance, angle phase, capacitance, and resistance values, in the frequency range of 0.01 Hz to 3.2 MHz, using the Impedance Analyzer Solartron 1260 (Solartron Analytical, AMETEK scientific instruments, Berwyn, PA, USA). For these measurements, a voltage amplitude of 1 V was applied to the terminals of interdigitated electrodes deposited on glass substrates. All measurements were temperature controlled using a thermal bath, and carried out at a constant temperature of 25.0 ± 0.2 °C. The electrical measurements in direct current mode were obtained through the characteristic current curve as a function of applied voltage, I (V). In this method, the current circulating in the device is measured when a certain voltage is applied to the device electrodes, using a programmable DC power supply (Rigol DP811A, Telonic Instruments, Berkshire, UK). For each voltage applied to the film, the respective current value was measured, through which it was possible to obtain the I (V) device characteristics. These measurements were also carried out under a spotlight simulating solar radiation, for which a halogen visible light bulb was used.

## 3. Results

### 3.1. UV-Visible Characterization of GO Aqueous Solution and Cast Films

To characterize the electronic structure of GO and quantify the adsorbed amounts in thin films, the UV visible spectrum of GO aqueous solutions with concentrations of 3 and 9 μM were measured (Figure 1a). The absorbance spectra presented two bands, in agreement with the literature [29]: one centered at 229 nm (attributed to π-π* electronic transitions at the C-C and C=C bonds of the aromatic rings), and the other located at 306 nm (related to n-π* transitions of the C=O bond).

From the UV-visible spectra of GO solutions obtained at different concentrations, it was possible to estimate the absorption coefficients (ε) using the Beer-Lambert Law, by plotting the absorbance at different wavelengths as a function of concentration, as shown in the inset plot of Figure 1a. The optical path length of the cuvette used for the measurements was 1 cm. The absorption coefficient values of the principal absorption peaks were 3.60 ± 0.09, 4.2 ± 0.2 and 1.80 ± 0.07 m^2^g^−1^, calculated from measurements at wavelengths of 200, 229, and 306 nm, respectively.

Graphene oxide cast films deposited onto CaF_2_ substrates were also characterized in the VUV wavelength region (Figure 1b). To determine the wavelengths of the absorption bands in this region, Gaussian curves were fitted to experimental data. GO VUV spectra presented four bands centered at the following wavelengths: 146 ± 4, 210 ± 20, 247.3 ± 0.2 and 299 ± 4 nm. The band located at 146 nm is related to transitions of π-π* type of the carbonyl group (aldehydes and ketones) [30]. The peak located at 210 nm is associated with a transition of π-π* type of the aromatic ring (phenol) [23] or to an n-π* transition of the carboxylic acid group. The maximum absorption band, located at 247.3 nm, can be associated with transitions of π-π* type of the aromatic ring (benzene) [30]. The band at 299 nm corresponds to an n-π* transition of the carbonyl group [30]. Table 1 shows the positions of the absorption bands detected in the VUV spectrum of the GO cast film, as well as the respective FWHM and the assignments of the electronic transitions involved.

### 3.2. Buildup of GO Based Layer-by-Layer Films

The buildup of PAH and GO LbL films as a function of the number of bilayers is shown in Figure 2a, where the UV-visible absorption spectra are displayed as a function of the number of bilayers. From the analysis of the spectra, it can be seen that similar absorption peaks are exhibited as for GO, which is expected as PAH has no significant absorption in this spectral region [31]. Fitting the data with Gaussian functions revealed that there were no significant deviations of the PAH/GO film absorption bands with respect to those obtained for GO in aqueous solution, i.e., 229 and 306 nm. In the case of the (PAH/GO)_20_ LbL film, the band at 225 nm is blue shifted by about 4 nm, while the band at 305 nm is 1 nm blue shifted. These small deviations may be associated with the electrostatic interaction between the PAH functional groups with positive (NH^3+^) charges and negative GO (COOH^−^) charges [32]. In addition, these deviations may also be related to the formation of GO aggregates [33]. According to Ferreira et al. [34], the aggregation of molecules may give rise to shifts in the absorption band wavelengths, since the electrical properties presented by aggregated molecules are different from those found in isolated chains [24]. Red shifts are associated with J-type aggregates, and blue shifts with H-type aggregates. Table 1 summarizes the position of the absorption peaks, full widths at half height (FWHH) and their respective electronic transition assignments.

To determine the mass of GO adsorbed per unit of area onto quartz substrates, the absorbance at 227 nm and at 307 nm was plotted as a function of the number of bilayers. These curves are plotted in the insert of Figure 2a and clearly show that the adsorbed amount grows linearly with the number of bilayers. Fitting this data with a straight-line yields slope values of 0.090 ± 0.003 per layer (*R*^2^ = 0.990) and 0.041 ± 0.001 per layer (*R*^2^ = 0.993), for the bands at 227 nm and at 307 nm, respectively. Based on these results, the amount of GO adsorbed per unit area for each bilayer can be estimated and, as expected, the film thickness is proportional to the number of bilayers. Using the value of the absorption coefficient at 229 nm already calculated for GO of 4.2 ± 0.2 m^2^ g^−1^, one can estimate the adsorbed amount of GO per bilayer, using the Beer-Lambert law. The measured absorbance has to be divided by two, since the layers are adsorbed on both surfaces of the quartz substrates. This calculation yields a value of 10.7 ± 0.1 mg m^−2^ of GO adsorbed per unit area and per layer. Moreover, these results show that the thickness of these films can be controlled simply by changing the number of bilayers.

The characterization of the absorption of PAH/GO LbL films was also performed through VUV spectroscopy in the wavelength range of 125 to 340 nm. In Figure 2b, the VUV spectra of (PAH/GO) LbL films with different numbers of bilayers deposited onto CaF_2_ substrates are shown. In this wavelength region, it is possible to observe the contribution of PAH absorption bands. These spectra were deconvoluted into Gaussian component curves, taking into account the respective features, namely, the average of peak positions and widths at half heights and assignments, as listed in Table 1. The band centered at 149.2 nm is a π-π* transition of the carbonyl group associated with aldehydes and ketones [23]. Parameters for fitting to the low intensity peak at 178 nm could not be found, but can be identified as being associated with n_N_-3pa type electronic transitions of pairs from solitary nitrogen electrons to orbitals of the same type (amine) [20] and n-σ* transitions from the carbonyl group (aldehydes and ketones) [23]. The band at 196.1 nm corresponds to n_N_-3sa transitions [21] and π-π* transitions in the aromatic ring (phenol) [23] and/or to the n-π transitions of the carboxylic group. The band at 199.1 nm can be attributed to the π-π* transitions of the aromatic ring (benzene). The two bands located at 225.3 nm and 305.1 nm result from π-π* and n-π* transitions of the carbonyl group, respectively.

To investigate the effect of UV radiation, the GO films were irradiated at 140 nm for 13 h. The VUV spectra before and after irradiation did not exhibit any appreciable changes, and thus no degradation in its structure was taking place.

The growth of PAH/GO LbL films deposited on CaF_2_ was also examined through measurement of the VUV spectra. It should be noted that for these VUV measurements, sets of films of varying numbers of layers were pre-prepared onto different CaF_2_ substrates before measurement. The spectra were not measured after sequentially adding layers between measurements, as done for the UV-Vis measurements. The films prepared for VUV measurement exhibited growth which was close to being linear, as demonstrated by the inset plot of Figure 2b, which relates the measured absorbance at 127 nm, 224 nm and 304 nm with the increasing number of bilayers. However, the errors associated with the method of preparation of these films mean that the errors are higher. Fitting these data with a straight line passing through the origin yields slope values of 0.18 ± 0.01 (*R*^2^ = 0.97), 0.118 ± 0.006 (*R*^2^ = 0.98) and 0.068 ± 0.005 (*R*^2^ = 0.95) for the bands at 127 nm, 224 nm and 304 nm, respectively. The linearity exhibited indicates that approximately the same amount of material was being adsorbed per bilayer and that the sequential growth of the bilayers is uniform, even considering different samples and with different bilayers. Comparison was also made between the growth of the band around 226 nm observed in PAH/GO films deposited on quartz and CaF_2_, measured by UV-Vis and VUV spectroscopy. The results indicate that a CaF_2_ substrate adsorbs a greater amount of material in comparison with quartz.

SEM was used to estimate the thickness of the LbL films. Figure 2c shows an image of the cross section of a (PAH/GO)_20_ LbL film deposited onto FTO with an aluminum layer deposited on top by thermal evaporation: a FTO/(PAH/GO)_20_/Al layer configuration. For the calculation of thickness, distance measurements were made at several points in the cross-sectional image, followed by use of the expression $d_{\mathrm{SEM}}=\frac{d_{\mathrm{Obs}}}{\cos\;20{}^{\circ}}$, where $d_{\mathrm{SEM}}\;$is the real thickness, $d_{\mathrm{Obs}}\;$is the average of the measured thickness values, and cos(20°) takes into account that the sample has a slope of 70° in relation to the incident electron beam.

The cross-sectional image of the FTO/(PAH/GO)_20_/Al device (Figure 2c) allowed an estimate of 210 ± 20 nm to be made for the thickness of (PAH/GO)_20_ LbL films deposited on FTO substrates, which corresponds to a thickness of 10.5 ± 0.7 nm per bilayer. From the cross-sectional images, the layers of PAH/GO LbL films appear to be well organized on the observed scale. The Al electrode obtained by thermal evaporation, Figure 2c, exhibited an average thickness, indicated by *d*_obs_ in the figure, of approximately 270 ± 10 nm. For the Al electrodes deposited through sputtering, a mean thickness of 360 ± 10 nm was estimated. The SEM data also indicate that the devices developed had the desired structure, and also that aluminum does not diffuse into the LbL films during the preparation of electrodes by metallization procedures, and therefore does not compromise the device operation. Moreover, through the cross-sectional images, it is possible to conclude that the LbL films appear to be well structured. The analysis of the aluminum electrodes shows that (results not shown here), although rougher, the surfaces of the films deposited by sputtering are more uniform than those obtained by thermal evaporation. No visible irregularity was observed in any of the films, such as cracks or crevices, which could jeopardize the proper operation of the devices.

FTIR spectra of different PAH/GO films with different numbers of bilayers deposited on CaF_2_ are shown in Figure 2d. The obtained FTIR spectra show that the growth of the film is not homogeneous, most obviously in the band near 1600 cm^−1^, associated with stretching of the C=C bond of the aromatic ring and with C=O vibrations in the ketone (GO) groups, which can be indicative of GO aggregation. This non-homogeneity in growth can be seen through optical microscopy (OM) images presented in Figure 3, which show the images of 1, 2 and 20 bilayer PAH/GO LbL films. The OM images of (PAH/GO)_1_ and (PAH/GO)_2_ were taken with different scales and show non-homogeneous surfaces, with both indicating the formation of possible clusters of PAH/GO. In addition, the formation of GO stacks/piles and/or PAH/GO stacks/piles was observed, as previously observed in films with one GO layer [35]. Images of PAH/GO films also reveal an increase in the size of particles and/or agglomerates in the films with two bilayers, when compared to films with a single bilayer.

As the number of bilayers increases, the surface becomes more heterogeneous, as can be clearly seen from the (PAH/GO)_20_ LbL film images presented in Figure 3 (3a–3c). The surface of the film, besides exhibiting PAH/GO aggregate formation, also showed GO and/or PAH/GO stacks. These images corroborate the inhomogeneity observed in the growth of the films, which was evidenced by the FTIR spectra (Figure 2d). In addition, it should be mentioned that these spectra were obtained from different films with a different number of layers deposited on separate CaF_2_ substrates and not a sequence of spectra measured during the growth of a single film, as done during the acquisition of the data displayed in Figure 2a. From the FTIR spectra of PAH/GO LbL films shown in Figure 2d, it is possible to identify the absorption bands, of which band wavenumber positions and vibration mode assignments are listed in Table 2.

### 3.3. Surface Characterization

The analysis of the surface morphology of the PAH/GO LbL films deposited onto quartz was carried out using AFM. In addition to the analysis of the (PAH/GO)_20_ film surface, the surfaces of films with a single layer of PAH and of GO, and LbL films with one and two bilayers, (PAH/GO)_1_ and (PAH/GO)_2_, were also characterized to obtain information about the growth dynamics of PAH/GO LbL films. The AFM images of the topography (a) and the phase (b), observed in an area of 1 × 1 μm^2^ of a PAH layer and of PAH/GO, (PAH/GO)_2_ and (PAH/GO)_20_ LbL films surfaces, are shown in Figure 4. The surface of the PAH film is homogeneous, presenting many small particles, which may correspond to the formation of small PAH aggregates, with dimensions varying between 29 nm and 190 nm. The surface is still quite rough, with a mean square roughness (Sq) of 4.15 nm and 3.87 nm for 1 × 1 μm^2^ and 2 × 2 μm^2^ measured areas, respectively.

AFM topographic images for different sample areas of the GO layer show that the film surface appears to be homogeneous with a Sq of 3.09 nm. The surface is formed by GO agglomerates which are distributed evenly across the surface. The film morphology reveals the formation of GO fibers, which can be associated with GO stakes/piles, with approximate lengths between 260 nm and 770 nm, widths between 28 nm and 105 nm, and heights of ~18 nm, which are values close to those reported by Xuan Wang et al. [35]. These fibers can be formed due to the parallel overlapping/stacking of GO sheets, with the ends of GO being folded or rolled up during the film production process.

The topographic surface data of the PAH/GO film reveals a surface that is slightly rough, with Sq of 3.09 nm and 3.91 nm for scan areas of 2 × 2 μm^2^ and 1 × 1 μm^2^, respectively. The images show a decrease in the number of particles on the film surface of (PAH/GO)_2_ compared to the PAH layer. It is further apparent that the number of GO fibers increases in relation to those observed on the surface of the GO film (Figure 4a). It is also revealed by the phase images that the GO layer covers the PAH layer, since there is no great contrast.

Morphological characterization through AFM was also performed on (PAH/GO)_20_ LbL films prepared with adsorption times of 1 min (Figure 4(a4)) and 3 min (topographic images not shown here). The topographic data show that the (PAH/GO)_20_ LbL films surfaces are not uniform at the observed nanoscale, with decreasing roughness and an increase of the deposition time of the LbL films. The film, obtained with 1 min of adsorption time per layer, presents Sq of 5.74 nm and 5.76 nm for the scan areas of 2 × 2 μm^2^ and 1 × 1 μm^2^, respectively, while films prepared with 3 min of adsorption time shows Sq of 4.50 nm and 2.74 nm for 2 × 2 μm^2^ and 1 × 1 μm^2^, respectively. The AFM images of the films described above indicate GO fibers having approximate widths of 17–70 nm and 18–73 nm, with heights between 15–31 nm and 5–17 nm, for adsorption times of 1 and 3 min, respectively. Moreover, from the phase images, significant contrast can also be observed, and so it can be concluded that the GO molecules were covering the entire surface of the PAH. Since the GO molecules were covering the entire PAH surface, this confirms that the process of obtaining thin films by the LbL technique was successful, producing well-organized layered films, in accordance with the results observed through SEM images. Table 3 summarizes the main morphology results of the films produced, namely mean square roughness values (Sq), mean roughness (Sa), surface area (SA) and fiber size width and length (minimum and maximum), and height of the stacks. The maximum height of the films and the dimensions of the aggregates (minimum and maximum) are also presented in the same table.

The plot of the logarithm of roughness as a function of the logarithm of scanning length may be able to indicate the type of film growth [29,39]. As linear dependence can be observed for the single GO layer and for the (PAH/GO)_2_ film, with slopes of 0.30 ± 0.03 and 0.4 ± 0.1, respectively, this may indicate that the examined films present a fractal growth in accordance with other results of growth of LbL films reported in the literature [34].

The surfaces of cast films of pure GO and GO solution (10^−2^ M) were found to be uniform, with both surfaces showing GO fibers. The GO film prepared from 10^−2^ M solution also shows GO aggregates. In both cases, the performed characterization shows that the deposited films seem to completely cover the FTO surface. The morphology of PAH/GO films, with 1, 2 and 20 bilayers, were also explored. As the number of bilayers is seen to increase, the surfaces of the films become less uniform, presenting an increase of PAH/GO aggregates and GO fibers and/or PAH/GO fibers. The non-uniformity exhibited by the PAH/GO films confirms the FTIR results, which point to a quasi-linear growth. The analysis carried out on aluminum electrodes deposited by sputtering and thermal evaporation show a similar morphology to the films on which they were deposited.

### 3.4. Electrical Measurements

Impedance spectroscopy was used to characterize the electrical properties of the PAH/GO LbL films, deposited onto glass substrates with gold interdigitated electrodes and measured in air at ambient room conditions. In Figure 5a,b, the real impedance and loss tangent spectra versus frequency in the frequency range of 10 to 10^4^ Hz are shown. For comparison, the data for a layer of GO is also included. As the frequency increases, the real impedance decreases monotonically for both GO and (PAH/GO)_20_ films, exhibiting the same type of behavior, with GO films being more conductive than the LbL films. This is easily understood as being due to PAH insulator properties. In addition, the dissipation factor, or Tg delta spectra, show that GO conducts better and that (PAH/GO)_20_ can store more energy.

Due to the presence of GO fibers, and in the case of LbL film parallel layers of different materials, the thin films analyzed here cannot be considered uniform from the electrical point of view. One might expect that the electrical measurements can be explained by the superimposition of the conductivity of different granular species such as GO molecules, GO stacks, and inter molecules and/or stacks regions. It should also be noted that in the case of PAH/GO LbL films, layers of PAH were surrounding the GO molecules. Each of these species can be electrically defined as a resistor and capacitor in parallel, all of which are placed together in series, and therefore, the frequency dependent real part of impedance can be expressed by the equation [40]:(1)$$Z^{\prime }=\frac{R_{1}}{\left(1+{\left(\omega R_{1}C_{1}\right)}^{2}\right)}+\frac{R_{2}}{\left(1+{\left(\omega R_{2}C_{2}\right)}^{2}\right)}+\frac{R_{3}}{\left(1+{\left(\omega R_{3}C_{3}\right)}^{2}\right)},$$
where, $R_{1}$, $R_{2}$ and $R_{3}$ are the resistance and $C_{1}$, $C_{2}$ and $C_{3}$ are the capacitance in the different species, namely, individual GO molecules, boundaries between molecules and GO stacks, and GO stacks, respectively in both samples, and ω is the angular frequency. The results attained from the fitting of Equation (1) to real impedance data of both samples is listed in Table 4.

The I (V) characteristic curves of the GO cast film obtained from a 10^−2^ M solution, poured onto a FTO substrate, and a film of (PAH/GO)_20_, adsorbed on FTO with an aluminum electrode deposited on the outermost surface, are displayed in Figure 6a,b. The measurements were performed under ambient conditions of pressure, temperature, and light. From the graphs in Figure 6b, the results obtained under illumination with visible light from a halogen lamp can also be found. The I (V) characteristic curve of the FTO/GO/Al device (Figure 6a) presents a semiconductor behavior of the film, thus confirming the GO conduction features. The measurements show that for positive bias, the device shows an almost negligible hysteresis effect, whereas for the negative bias, the observed hysteresis was seen to be larger. This hysteresis effect is related to the degradation of GO in the presence of oxygen and/or humidity, as can be inferred from the UV-Vis measurements (Figure 6c). Organic materials can be damaged by UV radiation at the molecular level, and GO molecules only absorb photons with wavelengths below 400 nm; therefore, the effect of irradiation of PAH/GO LbL films at ambient room conditions was investigated to see if damage could be observed. Films of (PAH/GO)_20_ were irradiated with 254 nm wavelength light for 13 h, the result of which is seen in the UV-Vis spectra taken of the sample before and after being irradiated (Figure 6c). After the 13 h of irradiation at ambient room conditions, the band located at 307 nm was no longer visible. GO can be deoxidized and transformed into reduced GO (rGO) through irradiation with UV light, and the disappearance of the band at 307 nm is likely associated with the break of the C=O bonds. In contrast, films measured in the VUV and irradiated at 140 nm did not show any change to the electronic structure. These measurements were performed under vacuum, and therefore, this lack of damage is likely due to the absence of oxygen and water molecules.

A semiconductor type behavior was also observed for the case of FTO/(PAH/GO)_20_/Al film devices, under both ambient light and under illumination with a halogen light source (Figure 6b). The analysis carried out under illumination conditions also showed a slight change in the device conduction. These differences can be explained by the damage observed in the PAH/GO films when they were irradiated at 254 nm in the presence of oxygen and water molecules from air. In both cases (ambient light and halogen light source), the I (V) curve also reveals slight hysteresis, when the device is positively polarized. When the device is negatively polarized and subjected to illumination with a halogen light, it also has negligible hysteresis, while under ambient light, it shows the largest hysteresis. The inset of Figure 6b shows the I (V) curve under these same conditions, but with the negative currents transformed into positive ones and on a new scale. The symmetry of the data plotted in the inset of Figure 6b shows that, under ambient light conditions, the current is slightly higher for positive voltages, while for measurements observed under the halogen source, there is no difference between the currents. This symmetry may also mean that there is not much difference between the working functions of the electrodes used. These results are in accordance with the recent achievements on hybrid solar-cells based in both PAH/GO and TiO_2_ or ZnO films, in which the developed architectures exhibit a diode like behavior and a decrease in current in a low-humidity environment [41].

This system has been proven to be quite stable both structurally and in terms of chemical reactivity, where an excessive reactivity would prove counterproductive to potential applications. This prototypical system of poly (allylamine hydrochloride) and graphene oxide presented in this work lends itself to considerable diversity in terms of thickness, doping and consequently electronic (band gap, conductivity) properties, and the methodology and approach used here is directly transferable to other systems.

## 4. Conclusions

Growth of PAH/GO LbL films was characterized by UV-Vis, VUV and FTIR spectroscopies. Both GO and PAH/GO films show the two π-π* and n-π* GO bands, which in solution are found at 229 nm and 306 nm. Small shifts of these bands were observed in the films attributed to electrostatic interactions. These films grow linearly indicating that each bilayer contains the same amount of GO. The GO surface density was determined to be 10.7 mg m^−2^ on the (PAH/GO)_20_ LbL film. The GO VUV spectrum has four absorption bands assigned to π-π* and n-π* transitions in the aromatic ring (phenol) and of the carboxylic group, whereas PAH/GO films have seven bands originating from both molecules. When GO is adsorbed on LbL films, deviations in band positions indicate interactions between GO and polyelectrolyte molecules.

When irradiated at 254 nm, the films showed a change in the GO structure, mainly associated with the C=O bond breaking. This is in contrast to a lack of GO structure change for irradiation at 140 nm, explained by the absence of oxygen and/or water molecules in the vacuum environment needed for the low wavelength irradiation.

The GO thin films had uniform surfaces with mainly fibers. The PAH/GO films also showed fibers, and became less uniform with increasing numbers of bilayers. Both films achieved full FTO surface coverage. SEM measurements of (PAH/GO)_20_ films showed a thickness of 209 nm, corresponding to about 10.5 nm per bilayer. These films had a relatively uniform surface with little roughness, exhibiting PAH/GO aggregates and fibers. Adsorption time influences the amount of GO adsorbed and the shortest time presented rougher surfaces. Furthermore, the images confirmed that the GO layer was completely adsorbed into the PAH layer. Roughness characterization by AFM concluded the presence fractal growth behavior.

Electrical impedance measurements showed that, as expected, a cast GO film has better conduction compared to PAH/GO films. However, for both films, three different granular species seem to contribute to the conduction. The characteristic I (V) curves revealed the typical behavior of Zener-like diodes for both FTO/GO/Al and FTO/(PAH/GO)_20_/Al devices, but also small hysteresis associated with degradation of the films via oxygen and/or water from air. Additionally, under illumination with a visible light source, the device with the FTO/(PAH/GO)_20_/Al layer architecture showed a further slight change in conduction. This device also displayed symmetrical behavior for positive currents, which may indicate that the working functions of the electrodes are similar. This study shows that the configuration of organic solar cells obtained via the LbL technique can be promising for the capture and conversion of solar energy into electric energy.

## Figures and Tables

**Figure 1 nanomaterials-11-01556-f001:**
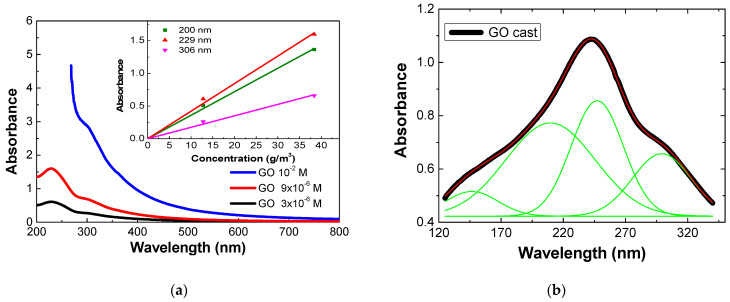
(**a**) UV-visible absorption spectra of aqueous GO solutions with concentrations of 3 × 10^−6^, 9 × 10^−6^ M and 10^−2^ M (for comparison). The inset shows the absorbance at different wavelengths as a function of the concentration of the GO aqueous solutions. (**b**) VUV absorption spectrum of a neat GO cast film deposited on a CaF_2_ substrate. Green lines are the obtained fit with Gaussian curves, and the red curve is the sum curve of all the fits.

**Figure 2 nanomaterials-11-01556-f002:**
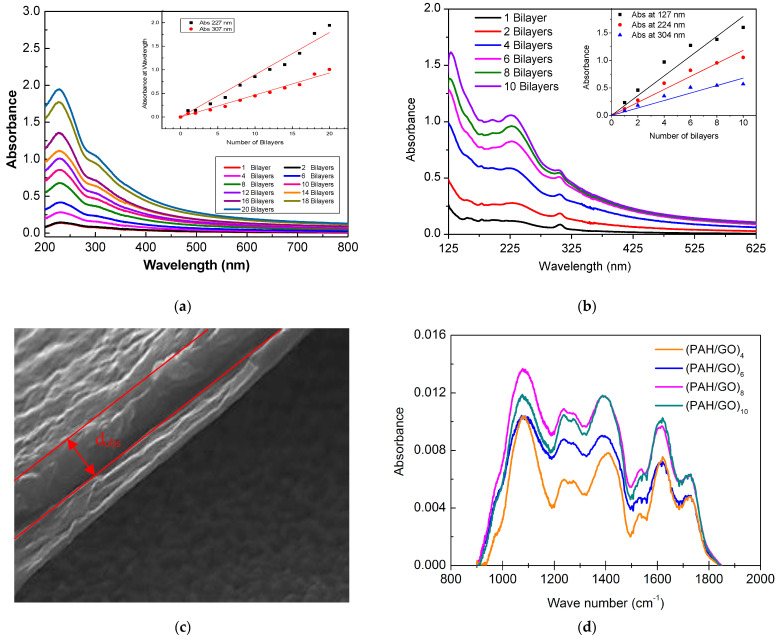
(**a**) UV-Vis absorption spectra for the different bilayers of a PAH/GO LbL film, prepared from aqueous solutions of PAH and GO with concentrations of 10^−2^ M. The inset shows the relation between the maximum absorbance at the bands, namely at 227 nm and 307 nm, as a function of the number of bilayers, n. (**b**) VUV absorption spectra for different bilayers of a PAH/GO LbL film deposited on CaF_2_ substrates, up to a maximum of 10 bilayers. The inset the absorbance of bands at 127 nm, 224 nm and 304 nm as a function of the number of bilayers, n. (**c**) SEM cross-sectional image of the device developed with the FTO/(PAH/GO)_20_/Al configuration with the Al layer with a thickness of 270 ± 10 nm, indicated by *d_obs_*, deposited by thermal evaporation. The (PAH/GO)_20_ film is the region between the Al layer and the dark region. (**d**) FTIR spectra of the growth of a PAH/GO film deposited on a CaF_2_ substrate, with the measurements performed after the 4th, 6th, 8th and 10th bilayers.

**Figure 3 nanomaterials-11-01556-f003:**
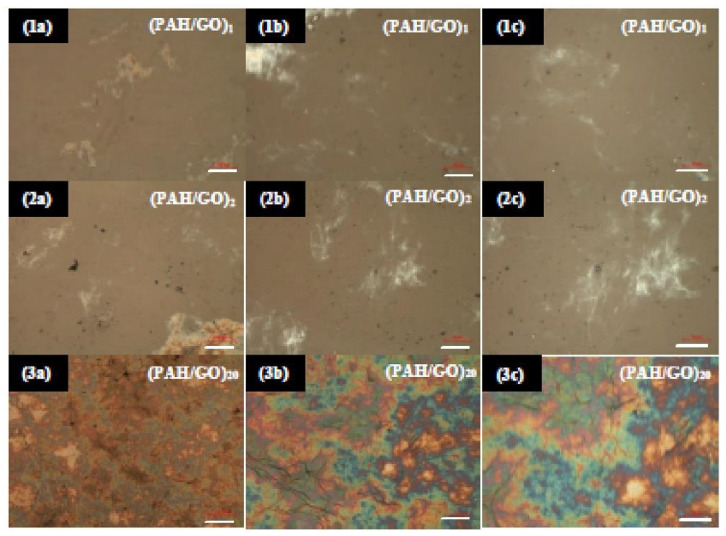
OM images, with different enlargements, of PAH/GO LbL film samples with 1 bilayer (**1a**–**1c**), 2 bilayers (**2a**–**2c**) and 20 bilayers (**3a**–**3c**) deposited on FTO. The white scale bars in the images correspond to 100 μm (**a**), 20 μm (**b**) and 10 μm (**c**).

**Figure 4 nanomaterials-11-01556-f004:**
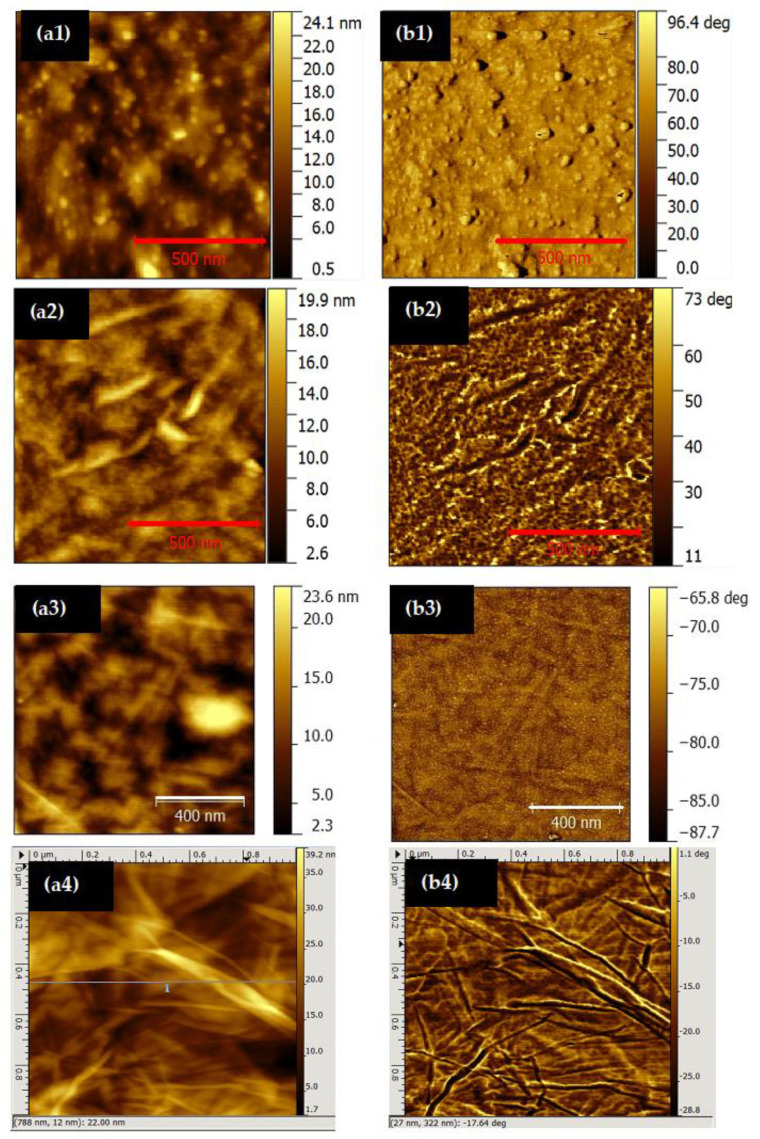
Topographic (**a**) and phase (**b**) images (1 × 1 μm^2^) of: (**1**) PAH layer; (**2**) PAH/GO bilayer; (**3**) (PAH/GO)_2_ LbL film; and (**4**) (PAH/GO)_20_ LbL film.

**Figure 5 nanomaterials-11-01556-f005:**
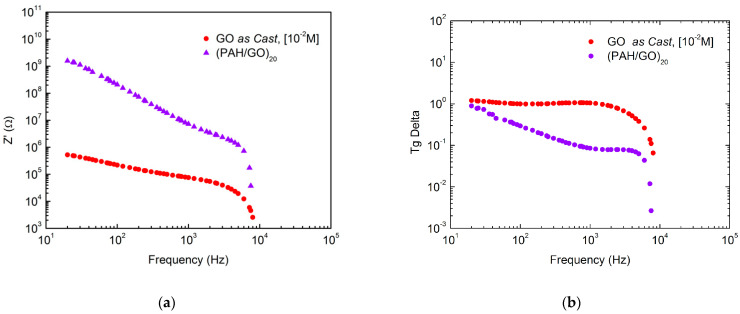
(**a**) Real impedance spectra and (**b**) tang delta spectra of PAH/GO LbL films and GO film in air at ambient conditions.

**Figure 6 nanomaterials-11-01556-f006:**
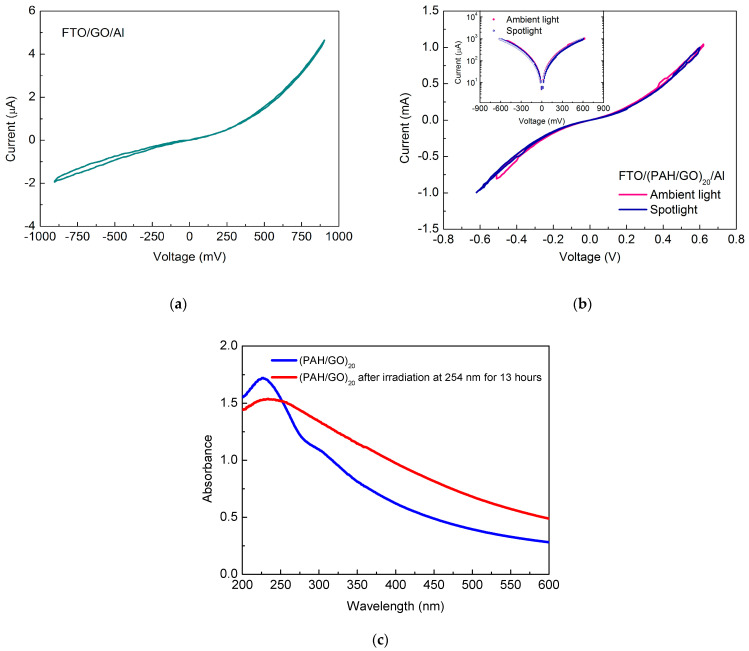
(**a**) Characteristics curves I (V) of the FTO/GO/Al device. (**b**) Characteristics curves I (V) of FTO/ (PAH/CO)_20_/AL, measured under ambient light and a halogen light source (spotlight), with the inset showing the same results presented as the larger figure (**b**), but now with change of scale and passage of negative to positive current. (**c**) UV-Vis spectra of (PAH/GO)20 LbL film before and after irradiation at 254 nm for 13 h.

**Table 1 nanomaterials-11-01556-t001:** Position of the absorption bands, full widths at half height (FWHH) and their respective assignments of the electronic transitions observed in the VUV spectra of GO deposited as cast and PAH/GO LbL films.

Films	Band Position (nm)	Band Energy (eV)	FWHH (nm)	Assignments to Electronic Transitions
GO Cast Films	146 ± 4	8.5 ± 0.3	50 ± 10	π-π* of the carbonyl groups (aldehydes and ketones)
	210 ± 20	5.9 ± 0.6	84	π-π* of the aromatic ring (phenol); n–π* of the carboxylic group
	247.3 ± 0.2	5.0 ± 0.8	49.5 ± 5.1	π-π* of the aromatic ring (benzene)
	299 ± 4	4.2 ± 0.1	55.4 ± 3.2	n-π* of carbonyl group
PAH/GO LbL films	127.8 ± 0.4	9.7	24.0 ± 0.8	----
	149.2 ± 2.3	8.3	33.2 ± 4.1	π-π* of the carbonyl groups (aldehydes and ketones)
	178 ± 1	7	---	n_N_-3pa of amine; π-π* of the aromatic ring (phenol); n-π* of the carboxylic group
	196.1 ± 0.7	6.3	9.3 ± 2.2	n_N_-3sa of amine; π-π* of the aromatic ring (phenol); n-π* of the carboxylic group
	199.1 ± 8.7	6.2	14.1 ± 6.9	π-π* of the aromatic ring (benzene);
	225.3 ± 0.1	5.5	81.2 ± 0.4	π-π* of carbonyl group
	305.1 ± 0.2	4.1	34.4 ± 0.7	n-π* of carbonyl group

**Table 2 nanomaterials-11-01556-t002:** Position of the peaks observed in the FTIR spectra of PAH/GO LbL films deposited on CaF_2_ substrates and the respective designations of the vibrational modes [18,22,36,37,38,39].

Peak Position (cm^−1^)	Peak Position in Literature (cm^−1^)	Assignments	
972	990.7	Vibrational modes of R-CH=CH_2_ groups [29].	PAH
1074.9	1088; 1048; 1080	Stretching C-N bonds of PAH, stretching vibrations C-O bonds of the C-O-C epoxide groups of GO and with vibrations C-OH groups of the COOH carboxyl groups of GO.	PAH, GO
1236.5 1272.6–1343.5	1230; 1227; 1226	Asymmetric stretching vibrations of the epoxide group C-O-C. Stretching of C-OH bonds.	GO
1548.6–1574.7	1500–1600	sp2 group hybridized at the C=C bonds in the vibrational plane (GO) and mainly with the torsion in the N-H oscillation and the vibration in the stretching C-N of the carbonate group (PAH).	
1619.6	1632; 1600–1650; 1628 and 1644; 1622	Stretching at the C=C bond of the aromatic ring, the C=O vibrations in the ketone (GO) groups, the vibrations due to torsion at the N-H bonds in the primary amines and with the stretching vibrations of the carbonate groups.	
1725.2	1723 and 1731; 1650–1750	C=O stretch of the carbonyl group; Vibrations on the COOH carboxyl group.	
2847.7	2847	Symmetrical stretching vibrations of C-H (CH_2_), (PAH).	PAH
2930.1	2921	Asymmetric stretching vibrations of C-H (CH_2_), respectively (PAH).	PAH
3085.9	3060	asymmetric N-H stretching vibrations of the -NH_3_^+^ group (PAH)	
3255.4	3250; 3050–3800;	O-H stretching at hydroxyl group notably phenol, C-OH (GO).	
3397.9–3587.2	3050–3800	O-H stretch in the hydroxyl group namely phenol, C-OH, which are also attributed to the COOH and H_2_O groups.	

**Table 3 nanomaterials-11-01556-t003:** Summary of mean square roughness values (Sq), mean roughness (Sa), surface area and fiber size and agglomerate’s area (AA), observed in the morphologically AFM characterized thin films.

	Area (μm^2^)	Sa (nm)	Sq (nm)	Stacks Width (nm)	Stacks Length (nm)	Height (nm)	SA (nm^2^)
PAH	4.08	3.03	3.81			31	
	1.02	2.68	3.41			27	18–190
GO	4.03	2.32	3.04			18	
	1.03	1.88	2.39	28–105	120–822	18	
	0.29	1.61	2.05				
(PAH/GO)_2_	4.05	2.65	3.34			21	
	1.12	2.50	3.12	43–135	222–670	19	23–50
	0.28	1.62	2.04				
(PAH/GO)_20_ ^#^	4.04	4.75	5.74				
	1.03	4.54	5.76	13.3–87	200–800	31	
(PAH/GO)_20_ *	4.24	3.63	4.50			26	
	1.01	2.17	2.74	11–62	281–564	17	

^#^ Adsorption time of 1 min. * Adsorption time of 3 min.

**Table 4 nanomaterials-11-01556-t004:** Attained values of the resistance and capacitance from fitting of Equation (1) to real impedance data of both samples.

	GO	PAH/GO
$R_{1}\left(\Omega \right)$	(4.49 ± 0.11) × 10^5^	(2.584 ± 0.072) × 10^9^
$2\pi R_{1}C_{1}$	0.0321 ± 0.0007	0.042 ± 0.002
$C_{1}\left(\mathrm{nF}\right)$	11.4	3 × 10^−3^
$R_{2}\left(\Omega \right)$	(1.25 ± 0.42) × 10^5^	(7.2 ± 0.3) × 10^7^
$2\pi R_{2}C_{2}$	0.0052 ± 0.0002	0.006 ± 0.002
$C_{2}\left(\mathrm{nF}\right)$	6.65	1.3 × 10^−2^
$R_{3}\left(\Omega \right)$	(8.29 ± 0.14) × 10^4^	(6.2 ± 1.6) × 10^6^
$2\pi R_{3}C_{3}$	0.00037 ± 0.00001	0.00043 ± 0.00009
$C_{3}\left(\mathrm{nF}\right)$	0.72	1.1 × 10^−2^
R^2^	0.99991	0.99943

## Data Availability

Not applicable.

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
