# Peer review of "Graphene Oxide Layer-by-Layer Films for Sensors and Devices"

_nanomaterials, 2021, doi:10.3390/nano11061556_

Round 1

Reviewer 1 Report

The author should answer following minor comments before getting a possible publication

  1. The formatting and grammatical errors in the article need to be checked carefully.
  2. The spelling mistake of the legend of X axis of the Fig.2a should be corrected.
  3. The figure numbers should be checked and corrected
  4. In Fig.4 Ib, and IIb , the scale bar is not visible clearly
  5. Conclusion should be compact and precise.
  6. The author should write purpose for each test in one/two sentences (in brief) before explaining the results of the characterization techniques. Therefore, the logic and organization of this part will be enhanced.

Author Response

Response to the Comments

We thank the reviewers for the comments and suggestions that we think are properly addressed in the present form of the manuscript. We think that our address to these comments, prompted by these reviews, improved its overall quality.

Reviewer 1:

The author should answer following minor comments before getting a possible publication

  1. The formatting and grammatical errors in the article need to be checked carefully.

Answer: The manuscript formatting and the English were carefully checked throughout.

  1. The spelling mistake of the legend of X axis of the Fig.2a should be corrected.

Answer: The spelling mistake of the legend of X axis of the Fig.2a and also for other figures were corrected.

  1. The figure numbers should be checked and corrected.

Answer: The figure numbers were checked and corrected.

  1. In Fig.4 Ib, and IIb , the scale bar is not visible clearly

Answer: The scale bars of Fig.4 Ib, and IIb were made visible.

  1. Conclusion should be compact and precise.

Answer: The conclusion was improved to become more compact and precise.

  1. The author should write purpose for each test in one/two sentences (in brief) before explaining the results of the characterization techniques. Therefore, the logic and organization of this part will be enhanced.

Answer:  A brief introduction was added to better introduce the experimental achievements.

Reviewer 2 Report

This manuscript reports on poly (allylamine hydrochloride) and graphene oxide (PAH/GO) thin films. The adsorption, the structural and electronic properties of these films were studied and discussed in detail. The proposed manuscript contributes an original work featuring interesting and prospective for device applications material system by employing adequate characterization efforts. The presented films including may even be seen as a prototypical in the field permitting variation by thickness, doping, etc. The authors succeed in a well-conceived and well-presented study of wide interest to the journal’s readership. In addition, the authors provided a good motivation behind embarking to such a study.

The analysis of the results is presented with high degree of clarity. The presentation of the results is concise yet convincing, easy to read/perceive and discussion insights are adequately described and critically emphasized.

Such work is really timely, while especially useful also because such thin films have high applicational relevance and are not well studied regarding their structure and electronic/electric properties. Also, the subject is seldom analyzed and discussed with such insight and clarity as in the present work. It is especially noteworthy the didactic way of presentation of the main characterization results.

In my opinion, the manuscript is bound to quickly attracting a quite significant research interest.

There are excellent figures too.

There are some minor aspects/questions to this, otherwise excellent manuscript, that need attention; thus, it is acceptable for publication after a minor revision:  

1: Introduction is well written. However, the authors completely miss that advanced modelling methods successfully guide and assist the synthesis of such carbon based thin and ultrathin films. For the broadness of understanding of structural and assembly/synthesis/surface chemistry issues on these and other similar thin film materials, it should be also mentioned that specially developed DFT theoretical approaches such as synthetic (growth) approach have applied to novel carbon-based thin films with inherent nanostructure [Journal of Physical Chemistry C 116 (2012) 21124; and Surface and Coatings Technology 206, (2011) 646].

2: When discussing the thin film systems for the present tests and possible applications, the authors should more clearly state/mention that this system is proven as quite stable both structurally and in terms of (lack of) excessive chemical reactivity which obviously would prove counterproductive to potential applications.

3: Generally speaking, it should be emphasized that the poly (allylamine hydrochloride) and graphene oxide system presented in this work can be seen as prototypical as to diversity of in terms of thickness, doping and consequently electronic (band gap, conductivity) properties, and the methodology and approach as used here is directly transferable to other such systems.

4: Spell-check and stylistic revision of the English of the paper are still necessary.

Author Response

Response to the Comments

We thank the reviewers for the comments and suggestions that we think are properly addressed in the present form of the manuscript. We think that our address to these comments, prompted by these reviews, improved its overall quality.

Reviewer 2

This manuscript reports on poly (allylamine hydrochloride) and graphene oxide (PAH/GO) thin films. The adsorption, the structural and electronic properties of these films were studied and discussed in detail. The proposed manuscript contributes an original work featuring interesting and prospective for device applications material system by employing adequate characterization efforts. The presented films including may even be seen as a prototypical in the field permitting variation by thickness, doping, etc. The authors succeed in a well-conceived and well-presented study of wide interest to the journal’s readership. In addition, the authors provided a good motivation behind embarking to such a study.

The analysis of the results is presented with high degree of clarity. The presentation of the results is concise yet convincing, easy to read/perceive and discussion insights are adequately described and critically emphasized.

Such work is really timely, while especially useful also because such thin films have high applicational relevance and are not well studied regarding their structure and electronic/electric properties. Also, the subject is seldom analyzed and discussed with such insight and clarity as in the present work. It is especially noteworthy the didactic way of presentation of the main characterization results.

In my opinion, the manuscript is bound to quickly attracting a quite significant research interest.

There are excellent figures too.

There are some minor aspects/questions to this, otherwise excellent manuscript, that need attention; thus, it is acceptable for publication after a minor revision:  

1: Introduction is well written. However, the authors completely miss that advanced modelling methods successfully guide and assist the synthesis of such carbon based thin and ultrathin films. For the broadness of understanding of structural and assembly/synthesis/surface chemistry issues on these and other similar thin film materials, it should be also mentioned that specially developed DFT theoretical approaches such as synthetic (growth) approach have applied to novel carbon-based thin films with inherent nanostructure [Journal of Physical Chemistry C 116 (2012) 21124; and Surface and Coatings Technology 206, (2011) 646].

Answer:  Thank you for the suggestion which is relevant. Sentences including this information have been included in the introduction as well as the suggested references.

2: When discussing the thin film systems for the present tests and possible applications, the authors should more clearly state/mention that this system is proven as quite stable both structurally and in terms of (lack of) excessive chemical reactivity which obviously would prove counterproductive to potential applications.

Answer:  Thank you for the suggestion. This detail has been added to the article.

3: Generally speaking, it should be emphasized that the poly (allylamine hydrochloride) and graphene oxide system presented in this work can be seen as prototypical as to diversity of in terms of thickness, doping and consequently electronic (band gap, conductivity) properties, and the methodology and approach as used here is directly transferable to other such systems.

Answer:  Thank you for the suggestion. This detail has been added to the article.

4: Spell-check and stylistic revision of the English of the paper are still necessary.

Answer:  The English was checked and revised throughout the manuscript.